# Validation of models using basic parameters to differentiate intestinal tuberculosis from Crohn's disease: A multicenter study from Asia

Julajak Limsrivilai[1,2☯], Choon Kin Lee[3☯], Piyapan Prueksapanich[4], Kamin Harinwan[5], Asawin Sudcharoen[1], Natcha Cheewasereechon[6], Satimai Aniwan[4], Pimsiri Sripongpan[6], Panu Wetwittayakhlang[6], Ananya Pongpaibul[7], Anapat Sanpavat[8], Nonthalee Pausawasdi[1], Phunchai Charatcharoenwitthaya[1], Peter D. R. Higgins[2], Siew Chien Ng[3]*

1 Division of Gastroenterology, Department of Medicine, Faculty of Medicine Siriraj Hospital, Mahidol University, Bangkok, Thailand, 2 Division of Gastroenterology, Department of Internal Medicine, University of Michigan, Ann Arbor, Michigan, United States of America, 3 Institute of Digestive Disease, Department of Medicine and Therapeutics, The Chinese University of Hong Kong, Sha Tin, New Territories, Hong Kong, 4 Division of Gastroenterology, Department of Medicine, Chulalongkorn University and King Chulalongkorn Memorial Hospital, Bangkok, Thailand, 5 Division of Gastroenterology and Hepatology, Department of Internal Medicine, Phramongkutklao Hospital, Bangkok, Thailand, 6 Division of Gastroenterology, Department of Internal Medicine, Songklanagarind Hospital, Prince of Songkla University, Songkla, Thailand, 7 Department of Pathology, Siriraj Hospital, Mahidol University, Bangkok, Thailand, 8 Department of Pathology, Chulalongkorn University and King Chulalongkorn Memorial Hospital, Bangkok, Thailand

☯ These authors contributed equally to this work.
* siewchienng@cuhk.edu.hk

**Data Availability Statement:** All relevant data are within the paper and its Supporting information files.

## Abstract

### Background

Data on external validation of models developed to distinguish Crohn's disease (CD) from intestinal tuberculosis (ITB) are limited. This study aimed to validate and compare models using clinical, endoscopic, and/or pathology findings to differentiate CD from ITB.

### Methods

Data from newly diagnosed ITB and CD patients were retrospectively collected from 5 centers located in Thailand or Hong Kong. The data was applied to Lee, *et al.*, Makharia, *et al.*, Jung, *et al.*, and Limsrivilai, *et al.* model.

### Results

Five hundred and thirty patients (383 CD, 147 ITB) with clinical and endoscopic data were included. The area under the receiver operating characteristic curve (AUROC) of Limsrivilai's clinical-endoscopy (CE) model was 0.853, which was comparable to the value of 0.862 in Jung's model (p = 0.52). Both models performed significantly better than Lee's endoscopy model (AUROC: 0.713, p<0.01). Pathology was available for review in 199 patients (116 CD, 83 ITB). When 3 modalities were combined, Limsrivilai's clinical-endoscopy-pathology

**Funding:** The authors received no specific funding for this work.

**Competing interests:** The authors have declared that no competing interests exist.

(CEP) model performed significantly better (AUROC: 0.887) than Limsrivilai's CE model (AUROC: 0.824, $p$ = 0.01), Jung's model (AUROC: 0.798, $p$ = 0.005) and Makharia's model (AUROC: 0.637, $p$<0.01). In 83 ITB patients, the rate of misdiagnosis with CD when used the proposed cutoff values in each original study was 9.6% for Limsrivilai's CEP, 15.7% for Jung's, and 66.3% for Makharia's model.

## Conclusions

Scoring systems with more parameters and diagnostic modalities performed better; however, application to clinical practice is still limited owing to high rate of misdiagnosis of ITB as CD. Models integrating more modalities such as imaging and serological tests are needed.

## Introduction

The incidence of inflammatory bowel disease or Crohn's disease (CD) has been increasing in Asia over the last few decades [1]. Moreover, differentiation of CD from intestinal tuberculosis (ITB) is difficult due to the low sensitivities of currently available diagnostic tests. The 5.3–37.5% sensitivity of acid fast bacilli (AFB) specimen staining [2–4], the 23%-46% sensitivity of mycobacterial culture [5, 6], and the 36.4–67.9% sensitivity of tissue polymerase chain reaction (PCR) [4, 5, 7–9] are all too low to confidently distinguish between these two conditions and exclude a diagnosis of ITB. The Asia-Pacific guideline recommends antituberculous therapy (ATT) for 8–12 weeks in patients with diagnostic uncertainty due to the of risk of disseminated tuberculosis if patients with ITB are misdiagnosed CD and they are prescribed immunosuppressive therapy [10]. However, treatment with ATT has many side effects and may delay treatment in patients with CD, and this may cause severe relapse and the development of complications [11]. In response, many studies were conducted to identify and classify characteristics that can help to distinguish between these two diseases. Those studies found that some clinical, endoscopy, pathology, radiology, and serology findings can help to improve diagnostic accuracy in these patients [2, 5, 7, 12–21]. However, no single diagnostic parameter can distinguish between CD and ITB. As a result, many models were developed that include various factors and modalities, and many of those models have been reported to have high performance. However, the number of studies performed to externally validate those models are limited.

Accordingly, the aim of this study was to validate and compare among scoring systems that use clinical, endoscopic, and/or pathology findings to differentiate CD from ITB in the same large multicenter cohort of patients. In this study, we focused on the models that integrate only basic parameters because the diagnostic methods and variables that are included in these models are widely available, which means that these diagnostic models can be used by physicians that work in limited-resource settings.

## Materials and methods

### Study design and data source

This multicenter retrospective cohort study included adult patients aged >18 years with a diagnosis of CD or ITB who were diagnosed and treated at four major GI centers in Thailand (Siriraj Hospital, King Chulalongkorn Memorial Hospital, Phramongkutklao Hospital, or Songklanakarin Hospital) from September 2004 to November 2018 and one major GI center

in Hong Kong (Prince of Wales Hospital, The Chinese University of Hong Kong) from January 2000 to November 2016. The data was collected in all centers between March 2017 and April 2019. The protocol for this study was approved by the institutional review board (IRB) of all participating centers including 1)The Joint Chinese University of Hong Kong—New Territories East Cluster Clinical Research Ethics Committee, 2)Siriraj Institutional Review Board, 3) The Institutional Review Board, Faculty of Medicine, Chulalongkorn University, 4) Institutional Review Board of Royal Thai Army Medical Department, and 5)Human Research Ethics Committee of Prince of Songkla University. The requirement to obtain written informed consent was waived due to the retrospective nature of this study.

Crohn's disease was diagnosed based on clinical, endoscopic, and pathology findings with clinical response to CD treatment which was defined based on physicians' notes in medical records that there was improvement of abdominal symptoms (such as abdominal pain, diarrhea, and bloody stool) and general well-being, in combination with the improvement of inflammatory biomarkers. The duration of at least six months of follow up was needed to confirm the clinical response. The criteria for diagnosis of ITB included any of the following: (i) presence of acid-fast bacilli or caseating granuloma in pathology specimens, (ii) tissue culture growing mycobacterial tuberculosis, (iii) presence of proven tuberculosis elsewhere in the body, or (iv) clinical and endoscopic response to ATT treatment without subsequent recurrence. A follow up endoscopy was performed at 2 to 6 months after initiation of treatment.

The clinical manifestations were manually reviewed in medical records. The pictures of endoscopic findings were reviewed by gastroenterologists of each center. The available pathologic slides were sent and reviewed by two pathologists from two centers. The gastroenterologists and pathologists were blinded to final diagnosis and any other predictive data. For pathological specimen slides, if the stain had faded out causing unclear images, repeat staining was performed on the slides.

## Model validation

Although several models which included clinical, endoscopy, pathology, cross-sectional imaging, and serology have been reported [22], only the models using clinical, endoscopic, and pathologic findings were included in this study. The models using only basic parameters have an advantage that these are standard investigations for the diagnosis of Crohn's disease or ITB in all countries including resource limited countries hence additional cost for investigations including interferon gamma release assay or cross sectional imagings are not required. The models that were validated in this study include the model by Lee, *et al.*, which includes 8 endoscopic findings [12]; the model by Makharia, *et al.*, which includes 4 parameters (2 clinical, 1 endoscopic, and 1 pathology) [7]; the model by Jung, *et al.*, which includes 7 parameters (4 clinical and 3 endoscopic) [23]; and, and the model by Limsrivilai, *et al.*, which includes 22 parameters (9 clinical, 8 endoscopic, and 5 pathology). A summary of each model is shown in Table 1. There were other two models which integrated clinical, endoscopic, and pathologic findings, but were not included in this study. The model by Yu *et al.* [14] integrated the presence of granuloma without specific characteristics, making it difficult to differentiate whether the granuloma was related to CD or ITB. The model by Li *et al.* [13] was also not included because of two reasons. First, this model integrated the presence of rodent-like ulcer without specific detail, and thus the interpretation was unclear. The other reason was the use of tuberculin skin test (TST) as one of the indicators for the diagnosis of ITB. Many patients in this cohort received BCG vaccine at birth, and it could cause a false positive TST. Therefore, physicians generally did not do TST in this setting. None of the patients in this cohort had this data available.

**Table 1. Summary of the models included in this study.**

| Authors | Country | Study design | Model |
|---|---|---|---|
| Lee YJ, *et al.* Endoscopy 2006 | Korea | Prospective CD 44, ITB 44 | Favors CD (+1/each): longitudinal ulcer, aphthous ulcer, cobblestone appearance, anorectal involvement<br>Favors ITB (-1/each): transverse ulcer, scars or pseudopolyps, a patulous ileocecal valve, involvement <4 segments<br>Final score: (+) Crohn's disease, 0: indeterminate, (-) ITB |
| Makharia, *et al.* Am J Gastroenterol 2010 | India | Prospective CD 53, ITB 53 (training) CD 20, ITB 20 (validation) | + 2.3 × weight loss– 2.1 × blood in stool– 2.5 × involvement of sigmoid colon– 2.1 × focally enhanced colitis + 7 |
| Jung Y, *et al.* Am J Gastroenterol 2016 | Korea | Retrospective CD 79, ITB 49 for training CD 79, ITB 49 for validation | $1/[1+e^{-(-4.423\ +\ 0.037*age\ +\ 2.226*sex\ -\ 2.203*diarrhea\ +\ 2.345*transvers\_ulcer\ -\ 1.911*longitudinal\_ulcer\ -\ 2.123*sigmoid\_colon\ +\ 5.606*pul\_tbc)}]$ |
| Limsrivilai, *et al.* Am J Gastroenterol 2017 | Meta-analysis of 38 studies comprising 2,117 CD and 1,589 ITB | | Model integrating 9 clinical, 8 endoscopic, 5 pathology, 5 CTE, and 1 IGRA bit.ly/ITBvsCD |

**Abbreviations**: CD, Crohn's disease; ITB, intestinal tuberculosis; CTE, computed tomography enterography; IGA, interferon gamma release assay

In this study, the model by Limsrivilai, *et al.* was alternatively named the ITBvsCD model, and it can be assessed via this link bit.ly/ITBvsCD. To compare the effectiveness of various combinations of parameters within the ITBvsCD model, we separated the diagnostic parameters, as follows: the clinical model (ITBvsCD-C), the endoscopy model (ITBvsCD-E), the clinical and endoscopy model (ITBvsCD-CE), and the clinical, endoscopy, and pathology model (ITBvsCD-CEP). The relative prevalence of ITB was required for this model calculation and the value of 0.28 was used. This relative prevalence of ITB was based on the total number of patients in this cohort, which included 427 CD and 163 ITB.

Data from patients who had available clinical and endoscopy data were applied to the Lee, *et al.*, Jung, *et al.*, and ITBvsCD (ITBvsCD-C, ITBvsCD-E, and ITBvsCD-CE) models. Data from patients who had clinical, endoscopy, and pathology data were applied to the ITBvsCD model (ITBvsCD-C, ITBvsCD-E, ITBvsCD-CE, and ITBvsCD-CEP), and the models from Jung, *et al.* and Makharia, *et al.* The performance of each model was assessed and compared to other models.

## Statistical analysis

Descriptive statistics were used to summarize patient characteristics. Continuous variables are expressed as median and range or mean ± standard deviation, and categorical variables are presented as number of subjects and percentage. Standard two-group comparison methods were used, including independent t-test or Wilcoxon rank-sum test for continuous data, and chi-square test or Fisher's exact test for categorical data. The performance of each model was assessed by area under the receiver operating characteristic curve (AUROC). DeLong test was employed to compare the performance of each model. The distribution of calculated probability of ITB from the ITBvsCD model is shown in box plots. The diagnostic performance of each model based on the proposed cutoff values from their original studies are reported as sensitivity, specificity, accuracy, positive and negative likelihood ratio (LR), positive predictive value (PPV), negative predictive value (NPV), and false-positive and false-negative rate. A two-tailed *p*-value of <0.05 was considered significant for all analyses. All analyses were performed using

SAS 9.4 (SAS Institute Inc., North Carolina, US) and R program version 3.2 (R Foundation for Statistical Computing, Vienna, Austria). Package OptimalCutpoints [24], pROC [25], epiR [26], and ggplot2 [27] were used.

## Results

Five hundred and ninety patients (427 CD and 163 ITB) were identified. Of those, 60 patients were excluded due to unavailable endoscopic data. The remaining 530 patients (383 CD and 147 ITB) with available clinical and endoscopic data were included in the analysis. Among ITB patients, 70 (47.6%) patients had pathological findings found either AFB or caseous granuloma, 35 (23.8%) patients had tissue culture growing mycobacterium tuberculosis, 36 (24.5%) patients had active tuberculosis elsewhere, and 29 (19.7%) patients were diagnosed based on response to empirical antituberculosis therapy. Nineteen (5%) of Crohn's disease patients had received antituberculosis therapy without response before the diagnosis of Crohn's disease was made. Demographic data, clinical manifestations, and endoscopic and pathology findings of study patients are shown in Table 2.

### Model validation and comparison

The data of 530 patients with available clinical and endoscopy data was used to validate the models integrating only clinical and endoscopy parameters. Of 530 patients, 199 patients (116 CD and 83 ITB) had pathology specimens available for review and those patients were included in the validation of all models.

**Validation of models integrating clinical and endoscopic parameters.** The data of 530 patients with available clinical and endoscopy data were applied to the ITBvsCD model and the models by Lee, *et al.* and Jung, *et al.* for model validation. In Lee's model, 143 patients obtained a score of zero, which reflects an indeterminate diagnosis. Among the remaining 387 patients, the sensitivity, specificity, and accuracy of diagnosis of ITB was 96%, 47%, and 61.2%, respectively. The AUROC was 0.713 (95% confidence interval [CI]: 0.677–0.748). Subgroup analysis in both study countries showed comparable model accuracy. The accuracy was 61.5% and 60.9% for Thai and Hong Kong cohort, respectively (S1 Table).

In the ITBvsCD model, the data was applied to the ITBvsCD-C, ITBvsCD-E, and ITBvsCD-CE models. The results of that analysis are shown in Fig 1. The AUROC of ITBvsCD-C and ITBvsCD-E was 0.756 (95% CI: 0.711–0.801) and 0.792 (95% CI: 0.752–0.831), respectively ($p = 0.21$). When the clinical and endoscopy were combined, the AUROC of the ITBvsCD-CE was 0.853 (95% CI: 0.817–0.888), which was significantly higher than both clinical alone and endoscopy alone (both $p < 0.01$).

When the data was applied to Jung's model, the AUROC was 0.862 (95% CI: 0.829–0.895). The performance between the ITBvsCD-CE model and Jung's model was not significantly different ($p = 0.52$), but both performed significantly better than Lee's model (both $p < 0.01$), which used only endoscopic findings. Subgroup analyses in the Thai and Hong Kong cohorts relative to the validation of the ITBvsCD model and Jung's model are shown in Table 3. Except for Jung's model, which performed significantly better in the Hong Kong cohort, the other comparisons between countries were non-significantly different.

**Validation of all models among the cohort of 199 patients with available clinical, endoscopy and pathology data.** The ITBvsCD model (ITBvsCD-C, ITBvsCD-E, ITBvsCD-CE, and ITBvsCD-CEP), and the models by Jung, *et al.* and Makharia, *et al.* were validated in this cohort.

Regarding the ITBvsCD model (as shown in Fig 2A), the model that integrated pathology findings further improved the performance compared to the ITBvsCD-CE model. The

**Table 2. Demographic, clinical, endoscopic, and pathology characteristics compared between Crohn's disease (CD) and intestinal tuberculosis (ITB).**

| | Total (n = 530) | CD (n = 383) | ITB (n = 147) | p |
|---|---|---|---|---|
| Age (years), (mean±SD) | 41.6±18.0 | 37.6±17.1 | 52.2±16.0 | **<0.01** |
| Male gender | 311 (58.7%) | 234 (61.1%) | 77 (52.4%) | 0.07 |
| *Clinical presentation, n (%)* | | | | |
| Duration of symptoms (months), median [IQR] | 6 [2–12] | 7 [3–15] | 3 [1–6] | **<0.01** |
| Abdominal pain | 352/520 (67.7%) | 265/374 (70.9%) | 87/146 (59.6%) | **0.01** |
| Diarrhea | 277/521 (53.2%) | 224/375 (58.6%) | 53/146 (36.3%) | **<0.01** |
| Hematochezia | 158/519 (30.4%) | 123/373 (33.0%) | 35/146 (24.0%) | **0.045** |
| Clinical gut obstruction | 39/519 (7.5%) | 28/373 (7.5%) | 11/146 (7.5%) | >0.99 |
| Fever | 102/518 (19.7%) | 50/372 (13.4%) | 52/146 (35.6%) | **<0.01** |
| Night sweats | 8/503 (1.6%) | 5/360 (1.4%) | 3/143 (2.1%) | 0.69 |
| Anemia | 206/515 (40%) | 143/369 (38.8%) | 63/146 (43.2%) | 0.36 |
| Weight loss | 250/517 (48.4%) | 169/371 (45.6%) | 81/146 (56.5%) | **0.04** |
| Perianal disease | 89/522 (17.1%) | 84/376 (22.3%) | 5/146 (3.4%) | **<0.01** |
| Extraintestinal manifestations | 47/520 (9.0%) | 41/374 (11.0%) | 6/146 (4.1%) | **0.01** |
| Lung involvement | 39/516 (7.6%) | 2/370 (0.5%) | 37/146 (25.3%) | **<0.01** |
| Ascites | 5/516 (1%) | 1/370 (0.27%) | 4/146 (2.7%) | **<0.01** |
| *Endoscopy, n (%)* | | | | |
| Longitudinal ulcer | 100 (18.9%) | 91 (23.8%) | 9 (6.1%) | **<0.01** |
| Cobblestone appearance | 48 (9.1%) | 48 (12.5%) | 0 (0.0%) | **<0.01** |
| Aphthous ulcer | 197 (37.2%) | 165 (43.1%) | 32 (21.8%) | **<0.01** |
| Transverse ulcer | 81 (15.3%) | 26 (6.8%) | 55 (37.4%) | **<0.01** |
| Patulous ileocecal valve* | 44/521 (8.5%) | 25/376 (6.6%) | 19/145 (13.1%) | **0.02** |
| Intestinal luminal narrowing | 86 (16.2%) | 73 (19.1%) | 13 (8.8%) | **<0.01** |
| Mucosal bridging | 7 (1.3%) | 7 (1.8%) | 0 (0.0%) | 0.20 |
| Pseudopolyps | 90 (17.0%) | 77 (20.1%) | 13 (8.8%) | **<0.01** |
| Segment involved** | | | | |
| Ileal involvement | 258/511 (50.5%) | 192/374 (51.3%) | 66/137 (48.2%) | 0.53 |
| Cecal involvement | 209/520 (40.2%) | 139/376 (37.0%) | 70/144 (48.6%) | **0.02** |
| Ascending colon involvement | 181/521 (34.7%) | 132/377 (35.0%) | 49/144 (34.0%) | 0.83 |
| Transverse colon involvement | 148/523 (28.3%) | 121/379 (31.9%) | 27/144 (18.8%) | **<0.01** |
| Descending colon involvement | 126/530 (23.8%) | 105/383 (27.4%) | 21/147 (14.3%) | **<0.01** |
| Sigmoid colon involvement | 150/530 (28.3%) | 138/383 (36.0%) | 12/147 (8.2%) | **<0.01** |
| Rectal involvement | 119/530 (22.5%) | 114/383 (29.8%) | 5/147 (3.4%) | **<0.01** |
| Less than 4 segments involvement | 415/530 (78.3%) | 287/383 (74.9%) | 128/147 (87.1%) | **<0.01** |
| *Pathology (n = 199), n (%)* | | (n = 116) | (n = 83) | |
| Presence of granuloma | 80 (40.2%) | 21 (18.0%) | 59 (71.1%) | **<0.01** |
| •Confluent granuloma | 43 (53.8%) | 5 (23.8%) | 38 (64.4%) | **<0.01** |
| •Large granuloma | 45 (56.3%) | 7 (33.3%) | 38 (64.4%) | **0.01** |
| •More than 5 granuloma per section | 18 (22.5%) | 1 (4.8%) | 17 (28.8%) | **0.03** |
| •Mucosal granuloma | 69 (86.3%) | 13 (61.9%) | 56 (94.9%) | **<0.01** |
| •Microgranuloma | 54 (67.5%) | 15 (71.4%) | 39 (66.1%) | 0.65 |
| •Cuffing lymphocytes around granuloma | 53 (67.1%) | 10 (47.6%) | 43 (74.1%) | **0.03** |
| Ulcer lined by histiocytes | 64 (32.2%) | 27 (23.3%) | 37 (44.6%) | **<0.01** |
| Disproportionate inflammation | 87 (43.7%) | 48 (41.4%) | 39 (47.0%) | 0.43 |

*(Continued)*

**Table 2.** (Continued)

|  | Total (n = 530) | CD (n = 383) | ITB (n = 147) | *p* |
|---|---|---|---|---|
| Focally enhanced colitis | 132 (66.3%) | 79 (68.1%) | 53 (63.9%) | 0.53 |

A *p*-value<0.05 indicates statistical significance

*Patulous ileocecal valve could not be evaluated in some patients because they had undergone hemicolectomy or flexible sigmoidoscopy

**Endoscopic findings could not be evaluated in some patients because they had undergone hemicolectomy, they had impassable stricture, they had undergone flexible sigmoidoscopy, or the terminal ileum was not accessed

**Abbreviations**: CD, Crohn's disease; ITB, intestinal tuberculosis; SD, standard deviation; IQR, interquartile range

AUROC significantly increased from 0.824 (95% CI: 0.766–0.881) to 0.887 (95% CI: 0.841–0.933) (*p* = 0.01). Fig 2B shows that the difference of the calculated probability of ITB between the patients with CD and ITB was most in ITBvsCD-CEP when compared to ITBvsCD-C, ITBvsCD-E and ITBvsCD-CE.

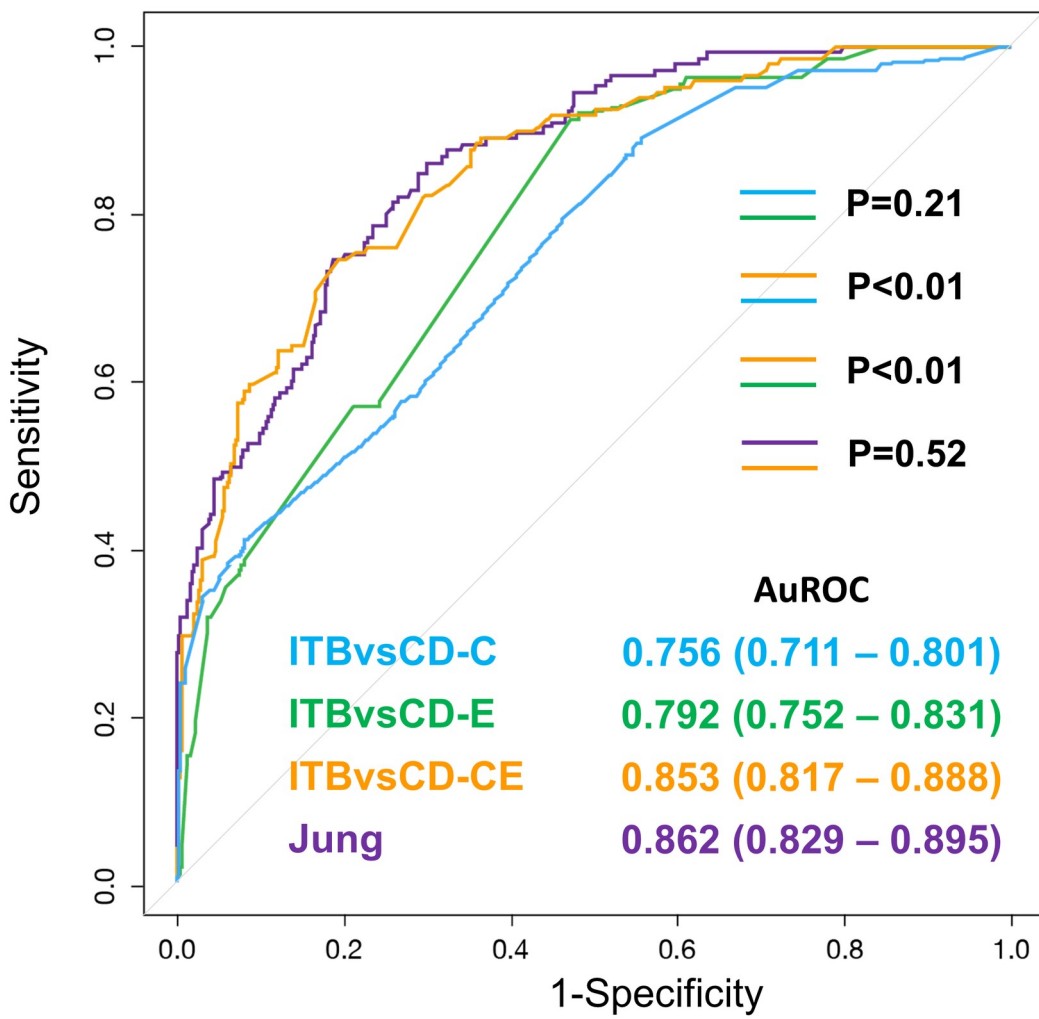

**Fig 1. Validation of the models integrating clinical and endoscopy in 530 patients who had clinical and endoscopic data.** Data shown as area under the receiver operating characteristic curve (AUROC), and a *p*-value <0.05 indicates statistical significance.

**Table 3. Subgroup analysis of validation of the model integrating clinical and endoscopic findings for each model compared among cohorts.**

| Model | AUROC | | | |
|---|---|---|---|---|
| | Total cohort (N = 530) | Thai cohort (n = 241) | Hong Kong cohort (n = 289) | *p* |
| **ITBvsCD-C** | 0.756 (0.711–0.801) | 0.744 (0.681–0.806) | 0.714 (0.637–0.791) | 0.55 |
| **ITBvsCD-E** | 0.792 (0.752–0.831) | 0.761 (0.703–0.819) | 0.778 (0.705–0.831) | 0.72 |
| **ITBvsCD-CE** | 0.853 (0.817–0.888) | 0.831 (0.781–0.882) | 0.827 (0.760–0.894) | 0.92 |
| **Jung's model** | 0.862 (0.829–0.895) | 0.810 (0.757–0.863) | 0.885 (0.833–0.938) | ***0.047*** |

A *p*-value<0.05 indicates statistical significance

**Abbreviation**: AUROC, area under the receiver operating characteristic curve; ITB, intestinal tuberculosis; CD, Crohn's disease; C, clinical findings; E, endoscopic findings; CE, clinical and endoscopic findings

Comparison among the ITBvsCD-CEP model, Jung's model, and Makharia's model is shown in Fig 3. The ITBvsCD-CEP model, which includes 22 variables from clinical, endoscopy, and pathology, performed significantly better than the model by Jung, *et al.* which includes 7 parameters from only clinical and endoscopy (AUROC: 0.798, 95% CI: 0.738–0.858), and the model by Makharia, *et al.*, which includes only 4 variables from clinical, endoscopy, and pathology (AUROC: 0.637, 95% CI: 0.561–0.713).

The sensitivity, specificity, accuracy, positive and negative LR, PPV, NPV, and false-positive and false-negative rates for diagnosis of intestinal tuberculosis for each model at the cutoff values proposed in the original studies, which included the calculated ITB probability of 20% in the ITBvsCD-CEP model [21], 0.35 in the model by Jung, *et al.* [23], and 5.1 in the model by Makharia, *et al.* [7], are summarized in Table 4. For the ITBvsCD-CEP model, other cutoff values were assessed to identify the most suitable cutoff value for use in clinical practice.

For the ITBvsCD-CEP model, the best cutoff value for differentiating CD from ITB was 59.47%. At this cutoff, the model diagnosed patients accurately in 82.9% of cases. However, the negative predictive value (NPV) was only 83%, which means there would have been 22 of 83 patients with ITB (26.5%) misdiagnosed as CD, and those patients would have received immunosuppressive agents. At the cutoff of 20%, which was reported to have an NPV of 100% in the original study [21], the NPV was 91% when applied to this cohort, which means that 8 of 83 ITB patients (9.6%) would have been treated as CD. To decrease the false-negative rate, the cutoff value was set to a lower value. To obtain an NPV of 95%, the cutoff value had to be set at 5%. However, at this cutoff value, as high as 55 of 116 CD patients (47.4%) would receive ATT without need.

For Jung's model, at the cutoff value of 0.35, the sensitivity, specificity, accuracy, and positive and negative predictive value were 84%, 56%, 67.8%, 58%, and 83%, respectively. The corresponding values in the original study were 98.0%, 92.4%, 95.2%, 88.9%, and 98.6%, respectively. For Makharia's model, at the cutoff value of 5.1, the sensitivity and specificity were 66% and 51%, respectively. The corresponding values in the original report were 90% and 60%, respectively, as shown in Table 4.

**Subgroup analysis in patients with intestinal tuberculosis who were diagnosed by response to empirical treatment with antituberculosis therapy.** Of 147 ITB, 29 patients had negative results for all diagnostic tests, and were diagnosed by response to empirical antituberculosis therapy. The models were applied to this group of patients. For ITBvsCD-CEP score, the median calculated ITB probability was 45.82% (range 2.81–99.96%). At the score cutoff value of 5%, 10%, and 20%, 2/29 (6.9%), 6/29 (20.7%), and 9/29 (31.0%%) ITB would be diagnosed with Crohn's disease, respectively. For Jung's score, the median score was 0.584 (range 0.009–0.999). At the cutoff value of 0.35, 7/29 (24.1%) ITB would be diagnosed with

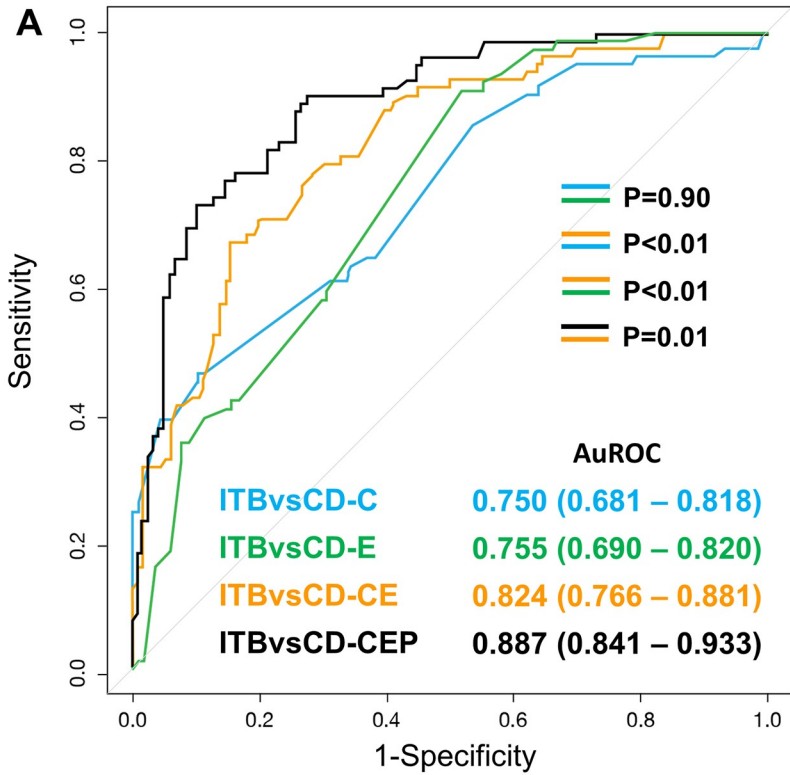

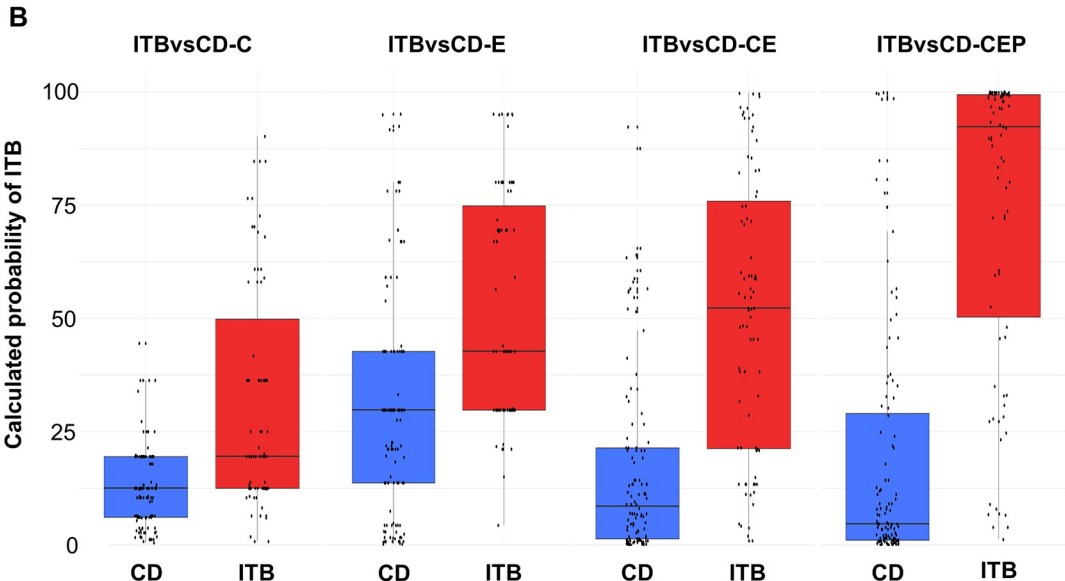

**Fig 2.** (A) Validation of the 4 ITBvsCD models in 199 patients who had clinical, endoscopic, and pathology data. Data shown as area under the receiver operating characteristic curve (AUROC), and a *p*-value <0.05 indicates statistical significance. (B) Distribution of the calculated probability of intestinal tuberculosis for each of the 4 ITBvsCD models. The bottom and top of each box represent the 25th and 75th percentiles, giving the interquartile range. The line through the box indicates the median, and the error bars indicate the 10th and 90th percentiles.

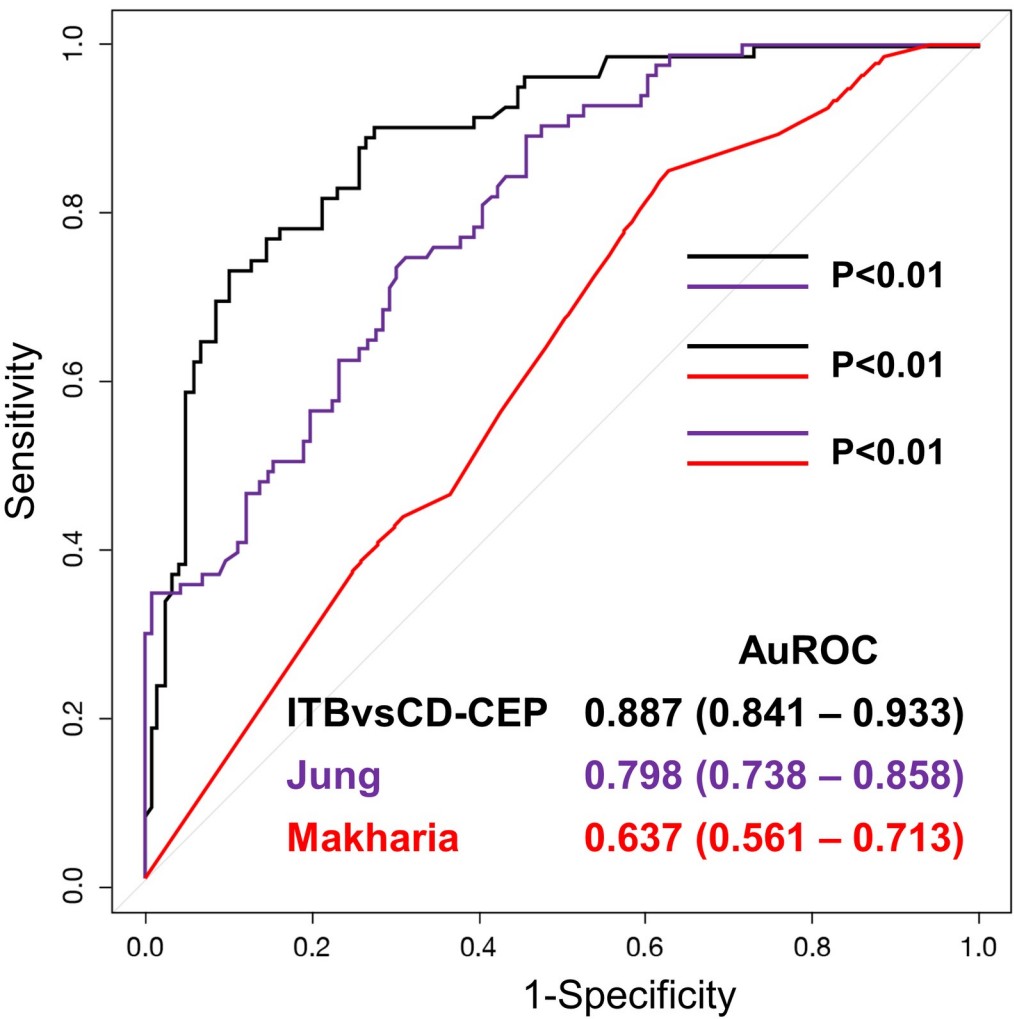

**Fig 3. Validation of the models integrating clinical, endoscopic, and pathology data in 199 patients who had clinical, endoscopic, and pathology data.** Data shown as area under the receiver operating characteristic curve (AUROC), and a *p*-value <0.05 indicates statistical significance.

Crohn's disease. For Makharia's score, 23 patients had available pathological specimens for review. The median score was 7.2 (range 2.6–9.3). At the cutoff value of 5.1, 8/23 (34.8%) of ITB would be diagnosed with Crohn's disease.

## Discussion

Many models for differentiating CD from ITB have been developed by many different research groups. Our study is the first to validate and compare these models in the same large multicenter cohort; however, our comparative analysis of models was intentionally limited to the models that include only clinical (alone and in combination), endoscopic (alone and in combination), and pathology (in combination only) parameters. Models that include radiographic and/or laboratory parameters were not included since these are considered advanced investigations and they are not available and/or affordable in many healthcare settings. We found that the ITBvsCD-CEP model, which includes clinical, endoscopy, and pathology, performed better than the models that include only clinical and endoscopy parameters

**Table 4. Performance of all models in diagnosis of intestinal tuberculosis in 199 patients with available clinical, endoscopic, and pathology findings.**

| Cutoff value | Sensitivity | Specificity | Accuracy | Positive LR | Negative LR | PPV | NPV | CD misdiagnosed as ITB | ITB misdiagnosed as CD |
|---|---|---|---|---|---|---|---|---|---|
| **ITBvsCD-CEP**: AUROC 0.887 (0.841–0.933) | | | | | | | | | |
| 59.47% | 74% | 90% | 83% | 7.10 | 0.30 | 84% | 83% | 12/116 | 22/83 |
| 20% | 90% | 71% | 79% | 3.08 | 0.14 | 69% | 91% | 34/116 | 8/83 |
| 10% | 90% | 63% | 74% | 2.44 | 0.15 | 64% | 90% | 43/116 | 8/83 |
| 5% | 96% | 53% | 71% | 2.03 | 0.07 | 59% | 95% | 55/116 | 3/83 |
| **ITBvsCD-CE**: AUROC 0.824 (0.766–0.881) | | | | | | | | | |
| 5% | 93% | 41% | 62% | 1.56 | 0.18 | 53% | 89% | 69/116 | 6/83 |
| **Model by Jung, *et al.***: AUROC 0.798 (0.738–0.858) | | | | | | | | | |
| 0.35 | 84% | 56% | 68% | 1.92 | 0.28 | 58% | 83% | 51/116 | 13/83 |
| **Model by Makharia, *et al.***: AUROC 0.637 (0.561–0.713) | | | | | | | | | |
| 5.1 | 66% | 51% | 57% | 1.35 | 0.66 | 49% | 68% | 59/116 | 55/83 |

**Abbreviations**: LR, likelihood ratio; PPV, positive predictive value; NPV, negative predictive value; ITB, intestinal tuberculosis; CD, Crohn's disease; AUROC, area under receiver operating characteristic curve

(ITBvsCD-CE and Jung's model). Furthermore, the models that include both clinical and endoscopy performed better than the clinical only model (ITBvsCD-C) and the endoscopy only models (ITBvsCD-E and Lee's model). We also found that the models that include clinical, endoscopic, and pathology parameters may not perform very well if the number of variables is low. For example, although it was found to be highly effective for differentiating ITB from CD in their cohort, the model by Makharia, *et al.*, which includes only 4 parameters, may not be generalizable to other populations that may have different characteristics. The concept of adding more variables to improve the diagnostic performance of a model is supported by two recent studies. First, Mao, *et al.* reported that adding significant CT enterography findings, including segmental involvement and comb sign, to the endoscopic score by Lee, *et al.* improved diagnostic accuracy from 66.7% to 95.2% [18]. Second, Bae, *et al.* integrated radiologic findings of pulmonary or small bowel involvement and serological tests, including IGRA and anti-Saccharomyces cerevisiae antibody (ASCA), into their original endoscopic score, and they found that the accuracy of their model improved from 81.2% to 96.3% [20].

Even though the AUROC for the ITBvsCD-CEP model is high at 0.887, there is still an important limitation relative to its application in clinical practice. Due to the risk of fatal complications if immunosuppressive agents are given to ITB patients who are misdiagnosed with CD, a tool needed is the one that can conclusively exclude ITB among patients with uncertain diagnosis. In the ITBvsCD-CEP model, at the best cutoff value of 59.4%, the false-negative rate was high at 26.5%, with 22 of 83 ITB patients being misdiagnosed. Although a lower threshold was set to lower the false-negative rate to even 5%, there would still be 3 ITB patients that would be misdiagnosed. Moreover, at this cutoff, about half of CD patients would receive ATT without actual need. Additionally, when applied all models to ITB patients with negative results of all diagnostic tests, and were diagnosed by empirical ATT, substantial numbers of patients in this group would have been misdiagnosed with Crohn's disease, emphasizing the limited value of these models. At the proposed cutoff value in the original studies, the percentage of misdiagnosis was 31.0% for ITBvsCD-CEP at the cutoff score of 20%, 24.1% for Jung' score at the cutoff of 0.35, and 34.8% for Makharia's score at the cutoff of 5.1. As such, the search for better models must continue. There are at least 2 ways to improve the ITBvsCD model. First, some basic parameters could also be beneficial. As shown in Table 2, most of the significant variables are concordant with the results of meta-analysis [21], and they have been

included in the ITBvsCD model. However, there are two significant findings that are not included in the ITBvsCD model—age and presenting duration. For age, the results of meta-analysis showed a trend of more advanced age in ITB, but this factor was not found to be statistically significant, so age was not included in the model. However, many studies published later that have not been included in meta-analysis, including studies by Jung, *et al.* [23], Bae, *et al.* [20], and He, *et al.* [28], reported that ITB patients were older than CD patients. Including these studies could have changed the result of meta-analysis. Regarding presenting duration, meta-analysis also showed presenting duration in CD to be longer than that in ITB. However, owing to the limitation of the ITBvsCD model, which can include only variables with dichotomous results, presenting duration could not be included in the model. The second way to improve the model is to add more advanced investigations, including CT enterography and serological tests, such as interferon gamma release assay. The ITBvsCD model was designed to include these variables, but the benefit of these advanced investigations needs to be proven in further studies.

The main strength of this study is that we included a large number of patients from two countries and from a total of 5 tertiary care centers. Furthermore, the results are quite comparable between the two countries, which suggests that our results may be generalizable to all Asian populations.

This study also has some limitations. First, due to the retrospective nature of this study, some clinical parameters might not have been available in medical records. Furthermore, the lack of mention of a particular finding in the official endoscopic and pathologic reports may not be meant as an absence of that finding. We have attempted to minimize this limitation by performing a review of pictures of endoscopic findings and specimen slides of pathologic findings by gastroenterologists and pathologists, respectively whereby an initial definitive diagnosis could not be made. For endoscopic findings, all available pictures of endoscopic findings of each patient were reviewed. For pathologic findings, 199 patients had slide specimens for review. The specimen slides of each patient were retrieved and reviewed by gastrointestinal pathologists. However, there may be some variations in the subjective evaluation of the findings of endoscopic lesions and pathology findings. Establishing clear definitions, or using central readers or advanced technology, such as artificial intelligence, for interpretation of the findings may be helpful. Second, models that include more than clinical, endoscopic, and pathology parameters, as shown in S2 Table [16–18, 29] and S3 Table [20, 28, 30, 31], were not able to be included in our study. Owing to the retrospective and real-life study design of this paper, not all subjects had undergone cross sectional imaging. Most of the imaging studies in our cohort were conventional CT abdomen and not CT enterography. Since the majority of the previous studies had used CT findings obtained from CT enterography for their models, extrapolating some of these findings to other models might not be accurate. Furthermore, many laboratory variables, including IGRA and ASCA were also not available. Further prospective studies aiming to validate other models which incorporated radiological imaging and serology will be necessary and this will form our future study. Third, the value of external model validation is limited by the small number of difficult to diagnose ITB. Only 29 (19.7%) patients had negative workup and were diagnosed by response to empirical anti-tuberculosis therapy.

## Conclusion

Scoring systems with more parameters and diagnostic modalities seemed to have significantly better ability to differentiate ITB from CD; however, application to clinical practice is still

limited owing to high rate of misdiagnosis of ITB as CD. Validation of models integrating more modalities such as imaging and serological tests is needed.

## Supporting information

**S1 Table. Validation of the score by Lee,** *et al.* **in our total cohort of 530 patients.**
(DOCX)

**S2 Table. Models integrating computed tomography enterography.**
(DOCX)

**S3 Table. Models integrating clinical, endoscopic, pathology, imaging, and laboratory findings.**
(DOCX)

**S1 Data.**
(CSV)

## Acknowledgments

The authors gratefully acknowledge Asst. Prof. Kevin P. Jones, Medical Research Manuscript Editor, Siriraj Medical Research Center (SiMR), Faculty of Medicine Siriraj Hospital, Mahidol University for language editing.

## Author Contributions

**Conceptualization:** Julajak Limsrivilai, Choon Kin Lee, Peter D. R. Higgins.

**Data curation:** Julajak Limsrivilai, Choon Kin Lee, Piyapan Prueksapanich, Kamin Harinwan, Asawin Sudcharoen, Natcha Cheewasereechon, Panu Wetwittayakhlang, Ananya Pongpaibul, Anapat Sanpavat.

**Formal analysis:** Julajak Limsrivilai.

**Investigation:** Julajak Limsrivilai.

**Methodology:** Julajak Limsrivilai, Choon Kin Lee, Peter D. R. Higgins.

**Resources:** Julajak Limsrivilai.

**Software:** Julajak Limsrivilai, Peter D. R. Higgins.

**Supervision:** Peter D. R. Higgins.

**Writing – original draft:** Julajak Limsrivilai.

**Writing – review & editing:** Satimai Aniwan, Pimsiri Sripongpan, Nonthalee Pausawasdi, Phunchai Charatcharoenwitthaya, Peter D. R. Higgins.

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
