## [Decision Letter · Decision Letter 0]

16 Jun 2020

PONE-D-20-14222

Validation of models using basic parameters to differentiate intestinal tuberculosis from Crohn's disease: A multicenter study from Asia

PLOS ONE

Dear Dr. Ng,

Thank you for submitting your manuscript to PLOS ONE. After careful consideration, we feel that it has merit but does not fully meet PLOS ONE’s publication criteria as it currently stands. Therefore, we invite you to submit a revised version of the manuscript that addresses the points raised during the review process.

I can understand that this MS has previously been evaluated for another journal and by chance, one reviewer that reject the paper has reviewed it again. However, in spite of his previously concerns he favor acceptance.

The other reviewer recommend rejection, primarily due to the retrospective design that are not clearly presented. In addition, it is not clear how these three models were chosen among several.

I can see some potentials in the MS, so I allow you to revise accordingly the reviewers raised issues.

We look forward to receiving your revised manuscript.

Kind regards,

Pal Bela Szecsi, M.D. D.M.Sci.

Academic Editor

PLOS ONE

Journal Requirements:

2. Please provide the date range when the medical data was originally collected, as well as the date(s) you accessed the data.

Reviewers' comments:

Reviewer's Responses to Questions

**Comments to the Author**

1. Is the manuscript technically sound, and do the data support the conclusions?

Reviewer #1: No

Reviewer #2: Yes

2. Has the statistical analysis been performed appropriately and rigorously? 

Reviewer #1: Yes

Reviewer #2: Yes

3. Have the authors made all data underlying the findings in their manuscript fully available?

Reviewer #1: Yes

Reviewer #2: Yes

4. Is the manuscript presented in an intelligible fashion and written in standard English?

Reviewer #1: Yes

Reviewer #2: Yes

5. Review Comments to the Author

Reviewer #1: The study is interesting, largely well written and has a large data set. However, there are important methodological issues which might affect the validity of the findings of the study.

Major Concerns

• It is surprising that only three models were chosen for evaluation. In a recent review (See Intestinal Res https://doi.org/10.5217/ir.2019.09142), there are a large number of models which have been reported. How did the authors choose these three models and ignore others. Also, radiology has a very important role and exclusion of radiological models is concerning.

• It is surprising that the authors give an impression that the model is validated in a prospective manner . See the wording in abstract “newly diagnosed ITB and CD patients”. It is in the methods that it becomes clear that this data has been validated in a retrospective cohort. The word retrospective should be mentioned in the abstract

• The major concern about this study is that this is a validation study on a retrospective cohort. Since a number of parameters would not be recorded the performance of a particular algorithm/model will not be truly evaluated. This is a single factor which compromises to a great extent the findings of the study.

• Also, in retrospective studies the lack of mention of a particular finding may be taken as absence of that finding. Eg. An endoscopist may ignore or not mention cobble-stoning unless the preformat mentions that this findings be reported OR endoscopist may ignore small/apthous ulcers when larger ulcers are present. This is a problem inherent in retrospective studies and therefore compromises the validity of findings.

• In some of the figures numbers don’t match: See night sweats, EIM in Table 2 ; Weight loss there is a gross mismatch. This gives an impression of carelessness in manuscript preparation and leaves the reviewer unsure whether to trust these findings

Minor Concerns

• It is incorrect to report that none of the models has been externally validated. The Limsrivilai model was created on basis of a Bayesian methodology using all published literature. This was then validated in a single center which is an external validatation (outside the dataset from which it was derived) .

• What is the definition of clinical response fo Crohn disease. Was CDAI used or were endoscopies done and SES-CD calculated

• One of the models also takes into account the underlying prevalence of ITB or CD. What was the prevalence chosen. One believes that since it is a cross country study the prevalence would differ

• How was response to ATT defined: was it endoscopic or clinical and what was the timing of response

Reviewer #2: This study looks at predictive models to differentiate between ITB and CD and looks at retrospective data sets to identify their reproducibility . It is a two center study and looks at three predictive models .

6. PLOS authors have the option to publish the peer review history of their article (what does this mean?). If published, this will include your full peer review and any attached files.

Reviewer #1: No

Reviewer #2: No

---

## [Author Response · Author response to Decision Letter 0]

14 Jul 2020

Dear Professor Pal Bela Szecsi, 

We would like to thank the editor and reviewers for examining our manuscript entitled, “Validation of models using basic parameters to differentiate intestinal tuberculosis from Crohn’s disease: A multicenter study from Asia” (PONE-D-20-14222). We have reviewed each of the reviewers’ comments and provided a point-by-point response as below. We are submitting our revised manuscript for consideration and hope you find this version acceptable for publication in PLOS ONE.

Thank you very much

Yours sincerely, 

Julajak Limsrivilai, M.D., MSc.

Choon Kin Lee, M.D.

Siew C Ng, M.D., Ph.D.

Reviewer #1: The study is interesting, largely well written and has a large data set. However, there are important methodological issues which might affect the validity of the findings of the study.

Major Concerns

• It is surprising that only three models were chosen for evaluation. In a recent review (See Intestinal Res https://doi.org/10.5217/ir.2019.09142), there are a large number of models which have been reported. How did the authors choose these three models and ignore others. Also, radiology has a very important role and exclusion of radiological models is concerning.

 Response: Thank you for the positive comments and for raising an important issue. We appreciate that several other models have been reported which included cross-sectional imaging and serology. Owing to the retrospective and real life study design of this paper, not all subjects had undergone cross sectional imaging. Most of the imaging studies in our cohort were conventional CT abdomen and not CT enterography. Since the majority of the previous studies had used CT findings obtained from CT enterography for their models, extrapolating some of these findings to other models might not be accurate. We have included these points in the limitations under discussion (page 19, line 351-358). In addition, the models using only clinical, endoscopic, and pathologic findings have an advantage that these are standard investigations for the diagnosis of Crohn’s disease or ITB in all countries including resource limited countries hence additional cost for investigations including interferon gamma release assay or cross sectional imagings are not required. Validation of their performance is also warranted. 

We have included details on how we selected these four models (Lee et al., Makharia et al., Jung et al., Limsrivilai et al.) in the Methods section and the section on model validation (page 6, line 113-118). We have also included under limitations that further prospective studies aiming to validate other models which incorporated radiological imaging will be necessary and this will form our future study.

• It is surprising that the authors give an impression that the model is validated in a prospective manner. See the wording in abstract “newly diagnosed ITB and CD patients”. It is in the methods that it becomes clear that this data has been validated in a retrospective cohort. The word retrospective should be mentioned in the abstract

Response: Thank you for the comments. We have added the word “retrospectively” in the abstract to better reflect our cohort.

• The major concern about this study is that this is a validation study on a retrospective cohort. Since a number of parameters would not be recorded the performance of a particular algorithm/model will not be truly evaluated. This is a single factor which compromises to a great extent the findings of the study.

Response: Thank you very much for the comments. We agree with the reviewer and have included this in the limitations of this study on page 18-19, line 338-346. 

• Also, in retrospective studies the lack of mention of a particular finding may be taken as absence of that finding. Eg. An endoscopist may ignore or not mention cobble-stoning unless the preformat mentions that this findings be reported OR endoscopist may ignore small/apthous ulcers when larger ulcers are present. This is a problem inherent in retrospective studies and therefore compromises the validity of findings.

Response: Thank you very much for the comments. We have attempted to minimize this limitation by performing a review of pictures of endoscopic findings and specimen slides of pathologic findings by gastroenterologists and pathologist, respectively whereby an initial definitive diagnosis could not be made. For endoscopic findings, all available pictures of endoscopic findings of each patient were reviewed. For pathologic findings, 199 patients had slide specimens for review. The specimen slides of each patient were retrieved and reviewed by gastrointestinal pathologists. If the stain had faded out, resulting in unclear images, repeat staining was performed on the slides. We have added this information in the Methods section on page 6, line 107-111 and also emphasize this limitation in the discussion on page 18-19, line 338-346. 

• In some of the figures numbers don’t match: See night sweats, EIM in Table 2 ; Weight loss there is a gross mismatch. This gives an impression of carelessness in manuscript preparation and leaves the reviewer unsure whether to trust these findings

Response: We appreciate the reviewer for the careful review and comments. We have corrected the typos and also rechecked other findings for accuracy.

Minor Concerns

• It is incorrect to report that none of the models has been externally validated. The Limsrivilai model was created on basis of a Bayesian methodology using all published literature. This was then validated in a single center which is an external validatation (outside the dataset from which it was derived).

Response: Thank you for the comments. We have revised the sentence “Models developed to distinguish Crohn’s disease (CD) from intestinal tuberculosis (ITB) have not been externally validated.” to “ Data on external validation of models developed to distinguish Crohn’s disease (CD) from intestinal tuberculosis (ITB) are limited.” in the abstract, and the sentence “However, none of these models have been externally validated and compared with the other models” to “However, the number of studies performed to externally validate those models are limited” in the first paragraph on page 4, line 73-74. 

• What is the definition of clinical response for Crohn disease. Was CDAI used or were endoscopies done and SES-CD calculated

Response: Clinical response was defined based on physicians’ notes in medical records that there was improvement of abdominal symptoms (such as abdominal pain, diarrhea, and bloody stool) and general well-being, in combination with the improvement of inflammatory biomarkers. As data in this study were collected retrospectively, neither CDAI nor SES-CD was routinely recorded. The duration of at least six months of follow up was needed to confirm the clinical response. We have added this detail in the Methods section on page 5, line 98-101. 

• One of the models also takes into account the underlying prevalence of ITB or CD. What was the prevalence chosen. One believes that since it is a cross country study the prevalence would differ

Response: Thank you for the comments, We used 0.28 for the prevalence of ITB and applied it to the ITB vs CD model. This number is based on the total number of patients in our cohort (427 CD and 163 ITB). We have added this detail in the Methods section on page 7, line 128-131.

• How was response to ATT defined: was it endoscopic or clinical and what was the timing of response.

Response: Among patients with negative results for the diagnosis of ITB, a diagnosis of ITB was based on empirical response to ATT. In this case, the response to ATT was defined as those having both a clinical and endoscopic response. A follow up endoscopy was performed at 2 to 6 months after initiation of treatment. This has been included in the methods section in page 6, line 105-106.

Reviewer #2: This study looks at predictive models to differentiate between ITB and CD and looks at retrospective data sets to identify their reproducibility. It is a two center study and looks at three predictive models.

Response: We thank the reviewer for the comments and summary of our paper. However, we do not see any suggestions. Please let us know if we miss something.

---

## [Decision Letter · Decision Letter 1]

16 Sep 2020

PONE-D-20-14222R1

Validation of models using basic parameters to differentiate intestinal tuberculosis from Crohn's disease: A multicenter study from Asia

PLOS ONE

Dear Dr. Ng,

Thank you for submitting your manuscript to PLOS ONE. After careful consideration, we feel that it has merit but does not fully meet PLOS ONE’s publication criteria as it currently stands. Therefore, we invite you to submit a revised version of the manuscript that addresses the points raised during the review process.

The manuscript has greatly improved, by especially one of the reviewers has issues that needs to be addressed.

We look forward to receiving your revised manuscript.

Kind regards,

Pal Bela Szecsi, M.D. D.M.Sci.

Academic Editor

PLOS ONE

Reviewers' comments:

Reviewer's Responses to Questions

**Comments to the Author**

1. If the authors have adequately addressed your comments raised in a previous round of review and you feel that this manuscript is now acceptable for publication, you may indicate that here to bypass the “Comments to the Author” section, enter your conflict of interest statement in the “Confidential to Editor” section, and submit your "Accept" recommendation.

Reviewer #2: All comments have been addressed

Reviewer #3: (No Response)

2. Is the manuscript technically sound, and do the data support the conclusions?

Reviewer #2: Yes

Reviewer #3: Partly

3. Has the statistical analysis been performed appropriately and rigorously? 

Reviewer #2: Yes

Reviewer #3: Yes

4. Have the authors made all data underlying the findings in their manuscript fully available?

Reviewer #2: Yes

Reviewer #3: Yes

5. Is the manuscript presented in an intelligible fashion and written in standard English?

Reviewer #2: Yes

Reviewer #3: Yes

6. Review Comments to the Author

Reviewer #2: The comments have addressed all points. I have no further comments . The authors have addressed all limitations.

Reviewer #3: The authors have done an interesting study comparing the existent models on differentiating CD and ITB. Though well presented, there are certain queries that need to be resolved-

1. The authors mention that they have compared the models which have utilized the only clinical, endoscopic and pathologic features so as to make it applicable in resource constraint countries. However, there are two other models by Yu et al (Digestion 2012;85:202-209.) and Li et al (Dig Dis Sci 2011;56:188-196.) which the authors have not included. Authors should discuss why these were excluded

2. One of these models (Li et al) includes tuberculin skin test which is easily available in resource constraint countries as well. Did the authors have data on TST in their cohort? If yes, then including this would add to the study.

3. The endoscopic and pathologic slides were reviewed by Gastroenterologist and Pathologists. Was central reading done or was it reviewed at different centers?

4. 29 of 147 ITB patients (20%) only had indeterminate diagnosis of ITB and rest had a definite diagnosis. Given the poor sensitivity of current diagnostic techniques for ITB, 80% definite diagnosis rate for ITB is rather surprising.

5. Authors should mention the proportion of patients who satisfied the different diagnostic criteria for ITB (e.g. positive AFB, culture, caseating necrosis, etc.)

6. Similarly among patients with CD, how many received ATT trial and how many were diagnosed upfront?

7. Among the comparison of pathologic findings, authors should also mention about the prevalence of caseation necrosis in either group

8. The real value of these models lies in predicting the diagnosis of ITB in patients with indeterminate TB. With only 29 such patients the study becomes grossly limited in its aim of external validation of these models. This point should be clearly highlighted in discussion.

7. PLOS authors have the option to publish the peer review history of their article (what does this mean?). If published, this will include your full peer review and any attached files.

Reviewer #2: No

Reviewer #3: No

---

## [Author Response · Author response to Decision Letter 1]

12 Oct 2020

Dear Professor Pal Bela Szecsi, 

We would like to thank the editor and reviewers again for examining our first revised manuscript entitled, “Validation of models using basic parameters to differentiate intestinal tuberculosis from Crohn’s disease: A multicenter study from Asia” (PONE-D-20-14222R1). We have reviewed each of the reviewers’ comments and provided a point-by-point response as below. We are submitting our revised manuscript for consideration and hope you find this version acceptable for publication in PLOS ONE.

Thank you very much

Yours sincerely, 

Julajak Limsrivilai, M.D., MSc.

Choon Kin Lee, M.D.

Siew C Ng, M.D., Ph.D.

Reviewer #2: The comments have addressed all points. I have no further comments. The authors have addressed all limitations.

Thank you very much. 

Reviewer #3: The authors have done an interesting study comparing the existent models on differentiating CD and ITB. Though well presented, there are certain queries that need to be resolved-

1. The authors mention that they have compared the models which have utilized the only clinical, endoscopic and pathologic features so as to make it applicable in resource constraint countries. However, there are two other models by Yu et al (Digestion 2012;85:202-209.) and Li et al (Dig Dis Sci 2011;56:188-196.) which the authors have not included. Authors should discuss why these were excluded

Thank you for raising this important issue. We were aware of these two models when we designed this study; however, we decided not to include them. The model by Yu et al. integrated the presence of granuloma without specific characteristics, making it difficult to differentiate whether the granuloma was related to CD or ITB. We also excluded the model by Li et al. because of two reasons. First, this model integrated the presence of rodent-like ulcer without specific detail, and thus the interpretation was unclear. The other reason was the use of tuberculin skin test (TST) as one of the indicators for the diagnosis of ITB. Many patients in our cohort received BCG vaccine at birth, and it could cause a false positive tuberculin skin test. Therefore, physicians generally did not do TST in this setting. None of the patients in our cohort had this data available. We have added this information in the section Materials and methods, subsection Model validation on page 6-7, line 124-133.

2. One of these models (Li et al) includes tuberculin skin test which is easily available in resource constraint countries as well. Did the authors have data on TST in their cohort? If yes, then including this would add to the study.

We agree that if tuberculin skin tests were available, it might have helped to distinguish ITB from CD. However, there is a concern about using TST in our patients as we mentioned above. 

3. The endoscopic and pathologic slides were reviewed by Gastroenterologist and Pathologists. Was central reading done or was it reviewed at different centers?

The endoscopic findings were reviewed by experts of each center. The available pathologic slides were sent and reviewed by two experts from two centers. We added this details in the methods section on page 6 line 107-110. We recognized interobserver disagreement, and thus stated this issue in the limitations of this study. We have added that using central readers could minimize the variation in the interpretation in the limitations of study on page 19, line 363 as well.

4. 29 of 147 ITB patients (20%) only had indeterminate diagnosis of ITB and rest had a definite diagnosis. Given the poor sensitivity of current diagnostic techniques for ITB, 80% definite diagnosis rate for ITB is rather surprising.

We agree with the reviewer and the detailed information about the diagnosis of ITB in this study is provided in the answer to question#5. The numbers of ITB patients whose diagnosis required response to empirical antituberculosis agents in this cohort is lower than we expected as well. 

5. Authors should mention the proportion of patients who satisfied the different diagnostic criteria for ITB (e.g. positive AFB, culture, caseating necrosis, etc.)

Among 147 ITB patients, 70 (47.6%) patients had pathological findings found either AFB or caseous granuloma, 35 (23.8%) patients had tissue culture growing mycobacterium tuberculosis, 36 (24.5%) patients had proven tuberculosis elsewhere, and 29 (19.7%) patients were diagnosed based on response to empirical antituberculosis therapy. We have added this information in the first paragraph of the results section on page 9 line 171-175. 

6. Similarly among patients with CD, how many received ATT trial and how many were diagnosed upfront?

Nineteen (5%) of Crohn’s disease patients had received anti-tuberculous therapy without response before the diagnosis of Crohn’s disease was made. We have added this information in the first paragraph of the results section on page 9 line 175-177. This proportion is quite low. The reason could be that this study was conducted in tertiary-referral centers where some physicians might have experience with Crohn’s disease more than general physicians, resulting in more confident to treat Crohn’s disease when all TB tests were negative. 

7. Among the comparison of pathologic findings, authors should also mention about the prevalence of caseation necrosis in either group. 

The prevalence of caseation in each group was not presented in the table 2 because the caseous necrosis was present only in patients with ITB. 

8. The real value of these models lies in predicting the diagnosis of ITB in patients with indeterminate TB. With only 29 such patients the study becomes grossly limited in its aim of external validation of these models. This point should be clearly highlighted in discussion.

Thank you for pointing out this issue. We do agree that the value of these models are limited to those with indeterminate diagnosis of ITB, which was quite a small number in this study, and we have added this limitation on page 20 line 373-376. Nonetheless, all diagnostic models were derived from patients with both indeterminate and definite ITB, and thus we opted to validate these models with similar patient population.

---

## [Decision Letter · Decision Letter 2]

11 Nov 2020

Validation of models using basic parameters to differentiate intestinal tuberculosis from Crohn's disease: A multicenter study from Asia

PONE-D-20-14222R2

Dear Dr. Ng,

We’re pleased to inform you that your manuscript has been judged scientifically suitable for publication and will be formally accepted for publication once it meets all outstanding technical requirements.

Kind regards,

Pal Bela Szecsi, M.D. D.M.Sci.

Academic Editor

PLOS ONE

Additional Editor Comments (optional):

Reviewers' comments:

Reviewer's Responses to Questions

**Comments to the Author**

1. If the authors have adequately addressed your comments raised in a previous round of review and you feel that this manuscript is now acceptable for publication, you may indicate that here to bypass the “Comments to the Author” section, enter your conflict of interest statement in the “Confidential to Editor” section, and submit your "Accept" recommendation.

Reviewer #3: All comments have been addressed

2. Is the manuscript technically sound, and do the data support the conclusions?

Reviewer #3: Yes

3. Has the statistical analysis been performed appropriately and rigorously? 

Reviewer #3: Yes

4. Have the authors made all data underlying the findings in their manuscript fully available?

Reviewer #3: Yes

5. Is the manuscript presented in an intelligible fashion and written in standard English?

Reviewer #3: Yes

6. Review Comments to the Author

Reviewer #3: The authors have satisfactorily answered all the queries.

7. PLOS authors have the option to publish the peer review history of their article (what does this mean?). If published, this will include your full peer review and any attached files.

Reviewer #3: No

---

## [Editor Report · Acceptance letter]

14 Nov 2020

PONE-D-20-14222R2 

Validation of models using basic parameters to differentiate intestinal tuberculosis from Crohn’s disease: A multicenter study from Asia 

Dear Dr. Ng:

I'm pleased to inform you that your manuscript has been deemed suitable for publication in PLOS ONE. Congratulations! Your manuscript is now with our production department. 

Kind regards, 

on behalf of

Dr. Pal Bela Szecsi 

Academic Editor

PLOS ONE